# Increased ACS Enzyme Dosage Causes Initiation of Climacteric Ethylene Production in Tomato

**DOI:** 10.3390/ijms231810788

**Published:** 2022-09-15

**Authors:** Haoting Chen, Songling Bai, Miyako Kusano, Hiroshi Ezura, Ning Wang

**Affiliations:** 1Graduate School of Life and Environmental Sciences, University of Tsukuba, Tsukuba 305-8572, Ibaraki, Japan; 2Department of Horticulture, Zhejiang University, Hangzhou 310058, China; 3RIKEN Center for Sustainable Resource Science, Yokohama 230-0045, Kanagawa, Japan; 4Tsukuba Plant Innovation Research Center, University of Tsukuba, Tsukuba 305-8572, Ibaraki, Japan

**Keywords:** wild tomato, ethylene biosynthesis, gene duplication, climacteric, fruit ripening

## Abstract

Fruits of wild tomato species show different ethylene-dependent ripening characteristics, such as variations in fruit color and whether they exhibit a climacteric or nonclimacteric ripening transition. 1-Aminocyclopropane-1-carboxylic acid (ACC) synthase (ACS) and ACC oxidase (ACO) are key enzymes in the ethylene biosynthetic pathway encoded by multigene families. Gene duplication is a primary driver of plant diversification and angiosperm evolution. Here, interspecific variations in the molecular regulation of ethylene biosynthesis and perception during fruit ripening in domesticated and wild tomatoes were investigated. Results showed that the activated *ACS* genes were increased in number in red-ripe tomato fruits than in green-ripe tomato fruits; therefore, elevated dosage of ACS enzyme promoted ripening ethylene production. Results showed that the expression of three *ACS* isogenes *ACS1A*, *ACS2*, and *ACS4*, which are involved in autocatalytic ethylene production, was higher in red-ripe tomato fruits than in green-ripe tomato fruits. Elevated *ACS* enzyme dosage promoted ethylene production, which corresponded to the climacteric response of red-ripe tomato fruits. The data suggest that autoinhibitory ethylene production is common to all tomato species, while autocatalytic ethylene production is specific to red-ripe species. The essential regulators *Non-ripening* (*NOR*) and *Ripening-Inhibitor* (*RIN*) have experienced gene activation and overlapped with increasing ACS enzyme dosage. These complex levels of transcript regulation link higher ethylene production with spatiotemporal modulation of gene expression in red-ripe tomato species. Taken together, this study shows that bursts in ethylene production that accompany fruit color changes in red-ripe tomatoes are likely to be an evolutionary adaptation for seed dispersal.

## 1. Introduction

Tomato (*Solanum* section Lycopersicon) includes all domesticated varieties and 12 wild relatives [1]. A great majority of wild tomato relatives are native to the Andes in the western side of South America from Ecuador to Northern Chile. The Andes region is characterized by a variety of climatic zones. These variable growth conditions gave rise to unique tomato phenotypes that are likely adaptive to the local habitat [2,3,4]. Fruits consumed by animals are carried over long distances to increase seed dispersal, which reduces parental competition and increases plant reproductive success [5]. Grumet and Herner [6] studied the ripening behavior of wild tomato relatives and grouped nine species into three categories. The first category includes the wild species *S. lycopersicum* var. *cerasiforme*, *S. pimpinellifolium*, and *S. galapagense*. Fruits of these species change color and show the typical climacteric ethylene production, which is a prominent feature of cultivated tomato fruits. The second category consists of *S. pennellii*, *S. habrochaites*, *S. chmielewskii*, and *S. parviflorum*. Species in this category have green fruits that ripen on the vine. Ethylene production is related to fruit softening in *S. chmielewskii* and *S. parviflorum*. However, there is no relationship between ethylene synthesis and fruit ripening in *S. pennellii* and *S. habrochaites*. *S. peruvianum* and *S. chilense* belong to the third category. Fruits of these species abscise prior to ripening and produce a trace amount of ethylene on the vine. The history of tomato domestication is considered as a “two-step” process in which modern cultivated tomato varieties were domesticated from *S*. *pimpinellifolium* that belong to the wild red-ripe species, after it gave rise to the intermediate species *S. lycopersicum* L. var. *cerasiforme* [7,8,9]. Red-ripe tomato species produce climacteric fruits that continue to ripen after harvest. Climacteric fruits show an increased respiratory rate and an ethylene production burst at the beginning of ripening and before the appearance of color [5,10].

The phytohormone ethylene is a key signaling molecule in the ripening of fruit [11]. 1-Aminocyclopropane-1-carboxylic acid (ACC) synthase (ACS) and ACC oxidase (ACO) are two key enzymes, and ACS is indicated as the rate-limiting enzyme in ethylene biosynthesis [12]. ACS and ACO are encoded by multigene families in higher plants. The tomato genome has at least nine *ACS* genes *(LEACS1A, LEACS1B*, and *LEACS2* to *8*) and five *ACO* genes (*LEACO1* to *5*) [13]. Multiple gene paralogs were created according to gene duplication events during the evolution of plants. Duplicated genes present a large potentiality for functional divergence and the new functions acquisition during plant evolution. Although most duplicated genes are lost over time [14], some are retained through neofunctionalization, subfunctionalization, or beneficial increases in gene dosage [15]. Duplicated genes create novel morphological, nutritional, and physiological phenotypes and exhibit changes in expression patterns from that of the original copy by altering the sequence of regulatory elements [16].

Two systems regulate ethylene production in climacteric fruits: autoinhibitory ethylene production called system 1 and autocatalytic ethylene production called system 2. System 1 is characterized by ethylene at basal levels in unripe fruits and vegetative tissues, which are regulated by *LEACS1A* and *LEACS6*. On the other hand, substantial increases in ethylene production during fruit ripening and flower senescence are regulated by *LEACS2* and *LEACS4* in system 2 [17]. Targeted inhibition of *LEACS2* [18] and *LEACS4* [19] gene expression reduced ethylene production and, thus, delayed fruit ripening. The transcription factors (TFs) *Ripening-Inhibitor* (*RIN*) and *Non-Ripening* (*NOR*) have been identified as essential regulator of tomato fruit ripening. [20,21]. *ACS* and *ACO* mRNA accumulation is inhibited in *RIN* and *NOR* knockout mutants, which delays fruit ripening [22,23]. In addition, RIN binding sites are identified in the *NOR* promoter region, and *NOR* promoter demethylation is required to activate gene expression during ripening [24].

In this study, interspecific variations in the molecular regulation of ethylene and carotenoids were investigated during fruit ripening of tomato in *S. lycopersicum* cv. Ailsa Craig, *S. pimpinellifolium*, *S. pennellii*, and *S. peruvianum*, which are species that exhibit the three ripening categories mentioned above. The expression levels of ethylene and carotenoid biosynthetic genes were correlated with ethylene production and coloration to identify ethylene-dependent divergence in fruits of red-ripe and green-ripe tomatoes. Promoter sequences were analyzed to identify regulatory elements responsible for modulating differential ethylene-related genes expression during ripening of fruit. Results of this study reveal that gene duplication events and modifications of upstream regulatory sequences in ethylene biosynthetic genes contributed to the diversification of domesticated tomato and their wild relatives. This research provides insights into the adaptive evolution of climacteric fruits and will facilitate the introgression of desirable traits from wild tomato relatives to cultivated varieties.

## 2. Results

### 2.1. Differences in Fruit Maturation and Ethylene Production during Ripening

The classic parameters of fruit skin color changes were documented to define the dynamics of fruit maturity stages in different tomato species. Fruit skin color was observed from 20 to 55 days after pollination (DAP) (Figure 1A). Fruits of the cultivated tomato, *S. lycopersicum*, var. M82 or Ailsa Craig, changed skin color from green (MG stage) to yellow (breaker, BR stage) at 40 ± 2 DAP. Fruits reached full maturity (red stage) at 50 ± 2 DAP. The BR stage of Moneymaker fruits was observed at 45 ± 2 DAP, and the red stage was observed at 50 ± 2 DAP. *S. pimpinellifolium* and S. *lycopersicum* var. *cerasiforme* fruits changed skin color from green to yellow (BR stage) at 38 ± 2 DAP and then reached the red stage at 48 ± 2 DAP. Fruits of *S. pennellii* and *S. peruvianum* remained green throughout the investigation period (Figure 1A).

Ethylene production rate was significantly different among fruits of red-ripe tomatoes and their green-ripe wild relatives (Figure 1B). In fruits of red-ripe tomatoes, ethylene production remained at baseline levels from 20 to 40 DAP. A rise in ethylene occurred at 40–50 DAP, and then declined thereafter. The peak ethylene production rates of M82, Ailsa Craig, and Moneymaker fruits were 0.004 ± 0.00107 μL/g/h, 0.0062 ± 0.00017 μL/g/h, and 0.0046 ± 0.00033 μL/g/h, respectively. Ethylene production of *S. pimpinellifolium* and *S. lycopersicum* var. *cerasiforme* fruits increased rapidly at 30 DAP. The peak ethylene production rate of *S. pimpinellifolium* fruits was 0.010 ± 0.00078 μL/g/h, which was recorded at 45 DAP, while that of *S. lycopersicum* var. *cerasiforme* fruits was 0.006 ± 0.00024 μL/g/h, which occurred at 40 DAP. *S. pennellii* and *S. peruvianum*, which have fruits that are green upon ripening, produced baseline levels of ethylene throughout fruit development and in over-ripe fruits (Appendix A). Taken together, the results show a correlation between ethylene production and fruit skin color changes.

### 2.2. Expression Profiling of Ethylene and Carotenoid Biosynthetic Genes

Accumulation of mRNA at different fruit developmental stages was observed in four representative tomato species, namely, *S. lycopersicum* cv. Ailsa Craig (a red-ripe cultivated tomato), *S. pimpinellifolium* (a red-ripe wild relative)*, S. pennellii*, and *S. peruvianum* (green-ripe wild relatives). Ripe fruits can be easily distinguished by skin color in red-ripe species, while time to fruit maturity is determined by observing changes in seed coat color in green-ripe species (Appendix A). *S. pennellii* and *S. peruvianum* changed seed coat color at 70 and 45 DAP, respectively. Thus, fruits at these stages were considered to be at maturity levels equivalent to the red stage of red-ripe species. *S. lycopersicum* cv. Ailsa Craig is one of the most widely studied cultivars in studies on tomato, which was used as a benchmark. Real-time quantitative PCR (qRT-PCR) showed that the highly expressed *ACS* genes were increased in number in red-ripe tomato fruits than in green-ripe tomato fruits (Figure 2A,B). For example, in the red-ripe cultivar, Ailsa Craig, the levels of *LEACS1A, LEACS2, LEACS4, LEACS6*, *LEACO1*, and *LEACO3* transcripts were relatively high at ripening. Furthermore, the expression level of *LEACS2* and *LEACS4* was upregulated in ripe fruits of *S. pimpinellifolium*. However, *LEACS1A* expression in *S. pimpinellifolium* fruits was 145-fold lower than that in Ailsa Craig fruits. In the green-ripe species, *S. pennellii*, *LEACO3* was the only gene that increased expression during ripening. Although *ACS* genes were expressed in *S. pennellii* fruits at the ripening stage, their levels were lower than those in Ailsa Craig fruits. Specifically, the expression of *LEACS2* in *S. pennellii* fruits was 2000-fold lower than that in Ailsa Craig fruits. The expression of *ACS* and *ACO* genes in *S. peruvianum* fruits was lower than that in Ailsa Craig fruits. Specifically, transcripts of *LEACS2* and *LEACS4* in *S. peruvianum* fruits were 970-fold and 32-fold lower, respectively, than those in Ailsa Craig fruits. On the other hand, *LEACS6* mRNA accumulated in immature fruits of *S. peruvianum*. Phytoene synthase (PSY) has been identified as the rate-limiting enzyme in carotenoid biosynthesis, and *PSY1* is transcriptionally regulated by ethylene in cultivated tomato. In, addition, z-carotene desaturase (ZDS) is a key enzyme for accumulation of lycopene, while lycopene ε-cyclase (LCYE) and lycopene β-cyclase (LCYB) produce cyclic carotenoids through lycopene breakdown [25]. Transcripts of *PSY1* showed 197-fold lower expression in *S. pennellii* and 48-fold lower expression in *S. peruvianum* compared to that in Ailsa Craig, respectively, and *ZDS* showed 120-fold lower expression in *S. peruvianum* and fivefold lower expression compared to that in Ailsa Craig (Figure 2C). Transcripts of *LCYE* showed 158-fold higher expression in *S. pennellii* and 250-fold higher expression in *S. peruvianum* compared to that in Ailsa Craig. Transcripts of *LCYE* were 1.8-fold higher in *S. pennellii* and 2.5-fold higher in *S. peruvianum* fruit. Expression of *RIN* and *NOR*, which encode ripening-related transcription factors, was higher in fruits of red-ripe species than in those of green-ripe species (Figure 2D).

### 2.3. Statistical Motif Analysis in Regulatory Sequences

Conservation of upstream regulatory sequences was analyzed for selected ethylene biosynthesis-related genes in cultivated tomato, *S. lycopersicum* var. *cerasiforme*, *S. pimpinellifolium*, and *S. pennellii*. The genes analyzed included *LEACS1A*, *LEACS2*, *LEACS4*, *LEACS6*, *LEACO1, LEACO3, RIN,* and *NOR* (Figure 3A and Appendix A). Promoter sequences of *LEACS6* and *RIN* were highly conserved among the four species. The regulatory sequence of *LEACS1A* had two more MYC-binding sites in cultivated tomato *S. lycopersicum* var. *cerasiforme* and *S. pennellii* than in *S. pimpinellifolium* (Figure 3A), and *S. pimpinellifolium* had one insertion located between −223 bp and −207 bp from *LEACS1A* transcription start site. Another insertion located between −168 bp and −123 bp was identified in the *LEACS1A* upstream sequence of red-ripe tomato species. The regulatory sequence of *LEACS2* had one more MYB and two fewer ERE cis-acting elements in *S. pennellii* compared to red-ripe tomato species. There were two insertions in the *LEACS2* upstream sequence of red-ripe tomato species and *S. pennellii*. The insertion located between motif 15 and motif 17 was identified in the *LEACS4* upstream sequence of cultivated tomato which was shared among red-ripe tomatoes. The sequence showed high similarity with Gypsy-36_BiGl-I retrotransposons (the Gypsy group of LTR retrotransposon family) (Appendix A). Screening of regulatory DNA elements revealed an increase in the number of MYB-binding sites in the *LEACS4* promoter of red-ripe tomatoes. The increased MYB-binding sites in the red-ripe tomatoes *LEACS4* promoter occurred in sequences with insertions located between motif 15 and motif 17, which contained two MYB-binding sites and one MYC-binding site (Figure 3A). Motif analysis also revealed that the promoter sequence located at −500 bp from the *LEACO1* transcription start site was highly conserved among red-ripe tomato species. Regulatory sequences of *NOR* were highly conserved among red-ripe tomatoes; however, the sequences located at −1100 to −2000 bp from *S*. *pennellii* were different (Appendix A). Conservation of upstream regulatory sequences was also analyzed for carotenoid biosynthesis-related genes. The *ZDS* upstream sequence showed two additional motifs (7 and 11) specific to the *S. pennellii* sequence and an additional motif-14 specific to red-ripe tomato species (Appendix A). The *LCYE* upstream sequence showed a large insertion (from motif 9 to 19) in *S. pennellii*, which revealed a number of MYB, MYC, W-box, and ERE cis-acting elements (Figure 3B). The insert sequence showed high similarity with MuDR-8_Stu transposons (the MuDR-type DNA transposon family) (Appendix A). The *LCYB* upstream sequence showed one more MYB-binding site in *S. pennellii*, but one fewer than in red-ripe species.

### 2.4. Exogenous Ethylene Response Assays

The triple response assay is used to evaluate how seedlings respond to exogenous ethylene. Seedlings typically exhibit short hypocotyls and roots, along with an exaggerated apical hook curvature, when exposed to ethylene. *S. pimpinellifolium*, *S. pennellii*, and *S. peruvianum* seedlings exposed to ethylene had hypocotyls and roots that were significantly shorter than those of controls and showed the formation of an exaggerated apical hook (Figure 4A,B).

Expression of *LEACS2* was positively induced by exogenous ethylene treatment in tomato fruits, except in *S. peruvianum*. *LEACS2* expression increased eight-, 20-, and twofold in Ailsa Craig, *S. pimpinellifolium*, and *S. pennellii*, respectively, when treated with exogenous ethylene. By contrast, exogenous ethylene inhibited *LEACS2* accumulation in *S. peruvianum* fruits (Figure 4C). *LEACS6* expression is negatively affected by exogenous ethylene treatment [17]. Here, *LEACS6* expression was four-, two-, three-, and fivefold lower in Ailsa Craig, *S. pimpinellifolium*, *S. pennellii*, and *S. peruvianum*, respectively, compared to the control (Figure 4C). Expression of *PSY1* was positively induced by exogenous ethylene treatment in tomato fruits, except in *S. peruvianum*. *PSY1* expression increased three-, 12-, and eightfold in Ailsa Craig, *S. pimpinellifolium*, and *S. pennellii*, respectively, when treated with exogenous ethylene (Figure 4D). Expression of *LCYE* was fivefold higher in *S. pennellii* and 3.5-fold lower in Ailsa Craig, when treated with exogenous ethylene (Figure 4D). *LCYB* expression was four-, four-, 1.5-, and threefold lower in Ailsa Craig, *S. pimpinellifolium*, *S. pennellii*, and *S. peruvianum*, respectively, when treated with exogenous ethylene. *ZDS* expression showed no significant difference.

### 2.5. Quantification of DNA Methylation in the NOR Promoter

Highly conserved elements at −1814 to −1404 bp of the *NOR* promoter were hypermethylated before ripening (Appendix A). Results of qAMP confirmed that the genome of cultivated tomato undergoes demethylation processes during ripening (Figure 5), which is consistent with the results of Zhong [24]. Cytosines of Ailsa Craig *NOR* promoter were hypermethylated at 20 DAP. Cytosine methylation levels then dropped rapidly at 30–45 DAP, while the *NOR* promoter remained in a demethylated state at 55 DAP. The *S. pimpinellifolium NOR* promoter showed a hypermethylated state at 20 DAP; however, it remained in a demethylated state at 30–55 DAP. The *S. pennellii NOR* promoter demethylated rapidly after 30 DAP, which was equivalent to the immature stage (i.e., turning stage at 70 DAP). The only species that was different with regard to *NOR* promoter methylation state was *S. peruvianum*. In this species, the *NOR* promoter remained in an unmethylated state at 20 DAP.

## 3. Discussion

Tomato includes thousands of varieties and 12 wild tomato species. Cultivated tomatoes are usually red when mature, which is the preferred color of consumers. Fruit color change is greatly involved in ethylene production. Autocatalytic ethylene production participates in ripening, which consists of mechanistic processes evolved for ground animal seed dispersal before red tomatoes were domesticated by humans. Although cultivated tomatoes can come in variety of colors, sizes, shapes, and flavor, there has been a sharp decline in genetic diversity due to domestication bottlenecks. Wild relatives reveal large genetic and phenotypic diversity which are desirable genetic resources for sustainable use of tomato breeding. Green-ripe wild relatives, such as *S. pennellii* and *S. peruvianum*, fail to show the typical climacteric ethylene production during fruit maturation. This research provides insights into the adaptive evolution of climacteric fruits and will facilitate the introgression of desirable traits from wild tomato relatives to cultivated varieties. The comparison between red-ripe and green-ripe tomatoes demonstrated a significant difference in the genetic regulation of ethylene and carotenoid biosynthesis. We found that increased ethylene production in red tomatoes is related to the enhanced dosage of ACS enzyme. Knowledge gained from the study of climacteric ethylene production will assist molecular manipulation, which is a rapid way to prevent adverse effects in wild relatives, including the non-climacteric feature.

### 3.1. Ethylene-Mediated Carotenoid Biosynthesis Is an Evolutionary Adaptation

Ripening in fleshy fruits is accompanied by changes in biochemistry, physiology, and structure, which attract seed-dispersing animals. Birds, in particular, are essential for plant reproductive success because they carry ingested seeds over long distances [4]. They tend to eat smaller fruits with bright colors, such as red or purple, as well as fruits without seed protection [26]. The gaseous plant hormone, ethylene, triggers physiological, biochemical, and molecular changes that lead to fruit ripening [27]. In this regard, inhibiting ethylene perception and signaling with 1-methylcyclopropene (1-MCP) or hydrogen sulfide (H_2_S) inhibits fruit coloration and softening [28,29]. This study showed that the green-ripe tomato species, *S. pennellii* and *S. peruvianum*, had low ethylene levels, which is a feature of non-climacteric fruits (Figure 1). By contrast, ethylene levels in red-ripe tomato species were significantly elevated at the onset of color change. The burst of ethylene production that accompanies fruit color change in *S. pimpinellifolium* is likely to be an evolutionary adaptation for seed dispersal. The bright color of red-ripe fruits is more attractive to tree dispersers, thus enabling increased opportunities for seeds to spread. Fruit color is specified by the type and balance of pigments in the pericarp. Tomato relatives produce fruits that are green, purple, yellow, or red [30]. Red and yellow colors in fruits are largely the result of carotenoids, while purple is caused by anthocyanin. Ethylene-driven activation of gene expression is a rate-limiting step in carotenoid biosynthesis. Ethylene-mediated ripening mechanisms of modern cultivated tomatoes are thought to be inherited from their wild ancestors. PSY1 and ZDS are key enzymes in the lycopene biosynthetic pathway, while LCYB and LCYE are required for the formation of α-carotene and β-carotene [25]. We demonstrated that red-ripe varieties triggered lycopene accumulation but lowered the formation of α-carotene and β-carotene. On the contrary, green-ripe wild relatives feature more accessible lycopene breakdown (Figure 2C). Interestingly, in *S*. *pennellii*, although the expression of *PSY1* was induced with exogenous ethylene, it circumvented any lycopene accumulation induced by the expression of *LCYE* (Figure 4D). *S. peruvianum* showed a high level of *LCYB* and *LCYE* expression upon ripening (Figure 2C), allowing lycopene breakdown. Other wild tomato relatives, such as *S. chilense* and *S. cheesmaniae*, synthesize anthocyanins in fruit epidermal tissue, leading to anthocyanin-spotted green-ripe fruits or fully purple fruits. Tomato wild species with purple fruits are grown in the high altitude and radiation-enriched Andean regions of South America [31]. Anthocyanins are induced by abiotic stresses, such as drought, high salinity, excess light, and cold. The purple and red color in wild tomato fruits could represent alternative evolutionary adaptations for attracting seed dispersers which are different from those in fruits exhibiting ethylene-mediated carotenoid biosynthesis.

### 3.2. Increased Dosage of ACS Enzyme Is Favored by Selection

The tomato genome contains multiple copies of genes that are closely related in structure and function. Early ancestors of cultivated tomato were probably polyploids that arose through two consecutive triplication events [32]. Therefore, tomatoes have a significant number of duplicated genes. Polyploidization increases plant diversity and is, therefore, considered a primary driver of angiosperm evolution. In the tomato genome, there are 11 *ACS* genes and six *ACO* genes [32]. These genes are differentially expressed at various stages of development and ripening. Tomato paralogous gene predictions indicate that *LEACS1A*, *LEACS2*, *LEACS4*, and two silenced *LEACS* genes belong to the ORTHOMCL532 cluster [32]. Alternatively, the amino-acid sequence that belongs to the protein encoded by *LEACS6* is not identical to that of the ORTHOMCL532 cluster. Previous studies on cultivated tomatoes revealed that *LEACS6* functions in wounding and plays a major role in developing fruits [33,34]. qRT-PCR revealed that *LEACS6* expression is highly conserved among red- and green-ripe tomato species, which indicated that *LEACS6* has an ancestral and essential function (Figure 2 and Appendix A). The expression of other duplicated genes, such as *LEACS1A*, *LEACS2*, and *LEACS4*, is temporally restricted during ripening. Results of this study indicate that duplicated ancestral genes followed by sub-functionalization contributed to tomato diversification.

Although duplicated genes can be retained without acquiring new functions, increased enzyme dosage could be beneficial because of elevated metabolic activity [15]. In red-ripe tomatoes, *S. pimpinellifolium* revealed higher ethylene production compared with cultivated tomatoes (Figure 1B). Although three ACS genes (*LEACS1A*, *LEACS2*, and *LEACS4*) were activated in cultivated tomato, the gene expression levels of two activated ACS genes (*LEACS2* and *LEACS4*) were significantly higher in *S. pimpinellifolium* (Figure 2A). Our survey indicates that ACS was the rate-limiting enzyme in ethylene biosynthesis. Clustering of *ACS* genes, such as *LEACS1A*, *LEACS2*, and *LEACS4*, which have similar functions in ethylene-mediated fruit ripening, is one example of metabolic benefits brought about by increased enzyme dosages. Transcript levels of *LEACS1A*, *LEACS2,* and *LEACS4* remained relatively low in mature fruits of *S. pennellii* and *S.*
*peruvianum* (Figure 2A and Figure 6), which correlated with lack burst of ethylene production and could explain the predominance of green coloration. In red-ripe tomatoes, the transient increase in *LEACS1A* transcripts was accompanied by elevated *LEACS4* expression at the beginning of ripening. *LEACS2* and *LEACS4* are in charge of the maintaining system 2 ethylene production among cultivated tomatoes [13]. According to the results presented here, activation of *LEACS2* and *LEACS4* gene expression is probably accompanied by the recruitment of activators to several enhancer sites in the upstream promoter region (Figure 3). It is plausible that tomato breeding led to the evolution of a complex autocatalytic system, in which *LEACS1A* activation triggers ripening-mediated ethylene biosynthesis in modern cultivated tomato cultivars.

### 3.3. Divergence in Transcriptional Regulation

Transposable elements (TEs) are considered one of the major drivers of genome evolution. Duplicated genes can arise adjacent to upstream transcriptional regulatory sites that alter their expression. Furthermore, TEs and repeat sequences can disrupt promoters of duplicated genes. In this study, relatively long insertions were found in the regulatory region of *ACS4* and *LCYE* (Figure 2). TEs represent a variable and high proportion of regulatory region; moreover, sequence motifs that potentially regulate the expression of *ACS4* and *LCYE* were identified in this study (Figure 3 and Appendix A). Additional MYB and MYC binding sites triggered by insertions in the *LEACS4* promoter were found in red-ripe tomatoes. *MYB* and *MYC* encode transcription factors (TFs), which regulate plant defense responses via jasmonate (JA) signaling [35] and ABA-dependent or -independent pathways [36]. A recent study suggested that *SlMYB70* overexpression delays fruit ripening via direct transcriptional repression of *LEACS2* and *LEACO3* [37]. Disruptive insertions within the promoter region of ethylene-related genes correlate with their function (Figure 2 and Figure 3), which could account for changes in ethylene but also carotenoid biosynthetic gene expression.

The TFs *NOR* and *RIN* are key regulators of ethylene biosynthesis during fruit ripening. Whole-genome bisulfite sequencing revealed that *RIN* binding sites are frequently localized in demethylated regions of numerous ripening gene promoters, including *NOR*. Moreover, *RIN* binding to promoters of ripening-related genes occurs simultaneously with demethylation [24]. The low levels of *NOR* expression in *S. pennellii* and *S. peruvianum* (Figure 2D) was accompanied by demethylated and unmethylated *NOR* promoter in *S. pennellii* and *S. peruvianum*, respectively (Figure 5). Therefore, a methylated promoter is not the cause of low *NOR* expression in green-ripe tomatoes. The involvement of NOR in ethylene biosynthesis appears to be an acquired mechanism to accelerate ethylene-dependent color changes during fruit ripening. In *S. pennellii*, the hypermethylated *NOR* promoter suggests that epigenetic regulation of *NOR* has been established. Although *S. pennellii* and *S. peruvianum* are green-ripe fruits, the genetic architecture of gene expression regulation is highly diversified among plants. Results presented here indicate that molecular pathways for *RIN*- and *NOR*-triggered ethylene production overlap with the increased ACS enzyme dosage (Figure 6). These complex levels of transcript regulation link higher ethylene production with spatiotemporal modulation of gene expression in red-ripe tomato species.

### 3.4. Ethylene Perception in Fruits

Although *LEACS6* transcript accumulation was associated with autoinhibition of ethylene production, *LEACS6* transcripts decreased rapidly at the onset of ripening [13]. Previous research demonstrated that *LEACS6* gene expression decreased after exogenous ethylene treatment [38]. In this study, *LEACS6* transcript abundance significantly declined after exogenous ethylene treatment in cultivated tomato and its three wild relatives (Figure 4C). Autocatalytic ethylene production participates in ripening and senescence, which are considered to be mechanistic processes that define climacteric fruits during evolution [5]. A major feature of autocatalytic ethylene synthesis in cultivated tomato is ethylene-inducible *ACS2* expression [10]. The observation that exogenous ethylene treatment significantly promoted *LEACS2* gene expression in Alisa Craig and *S. pimpinellifolium* was consistent with an autocatalytic ethylene biosynthetic pathway (Figure 4C). In the green-ripe wild tomato relative, *S. peruvianum*, *LEACS2* expression in fruits was low and was not upregulated by the treatment of exogenous ethylene (Figure 2A and Figure 4C). However, the observation of inhibited hypocotyl and root growth in the seedling triple response assay suggests the presence of functional ethylene receptors in *S. peruvianum (*Figure 4A,B). Inactivation of *LE**ACS2* coupled with low ethylene production could explain the lack of ethylene response in *S. peruvianum* fruits (Figure 6). By contrast, exogenous ethylene-inducible *LEACS2* occurred in fruits of *S*. *pennellii* (Figure 4C), which is another green-ripe wild tomato relative. Therefore, autocatalytic ethylene synthesis and minimal ethylene production during fruit ripening in *S*. *pennellii* are likely the result of lower *NOR* and *RIN* expression and a reduction in ACS enzyme dosage (Figure 6).

## 4. Materials and Methods

### 4.1. Plant Materials and Growth Conditions

Cultivated tomatoes (*S. lycopersicum* cv. M82, Moneymaker, and cv. Ailsa Craig) and their wild relatives (*S. pimpinellifolium*, *S. pennellii*, *S. peruvianum*, and *S*. *lycopersicum* var. *cerasiforme*) were provided by National BioResource Project Tomato (NBRP-Tomato) (https://tomatoma.nbrp.jp/ (accessed on 20 July 2022)). The plants were grown in a glasshouse located at the Tsukuba-Plant Innovation Research Center (T-PIRC) Farm, University of Tsukuba from April 2020 to January 2021 in a randomized scheme. Plants were fertilized once a week using HYPONeX (Hyponex, Japan). Flowers were marked on the date of pollination. Then, the fruits were sampled at 20 DAP to 55 DAP. After harvest, seeds and the fruit columella were removed immediately, ground in liquid nitrogen, and stored at −80 °C until use.

### 4.2. Ethylene Production in Fruits

Measurements of ethylene production were conducted on fruits that were harvested every 5 days for the duration of fruit development, which spanned 20–55 DAP. Fruits of cultivated tomatoes and their wild relatives were placed in 270 mL and 50 mL glass bottles, respectively. Bottles were left open for 1 h after placing the fruit inside. The air was then blown off to remove the effect of wound-induced ethylene production. Bottle caps were tightened, and bottles containing the fruit were incubated for 4 h at room temperate. A sample of air (1 mL) was taken with a medical syringe and injected into a Shimadzu 5890 series gas chromatograph equipped with a flame ionization detector. Ethylene concentration was normalized to reagent-grade ethylene standards with respect to fruit weight, volume, and time for incubation using the following equation:Ethylene concentration (μL/g fresh weight/h) = X × Y/Z × T × 1000, where X is the concentration (ppm), Y is the volume of the bottle (mL), Z is the fruit weight (g), and T is the time (h). 

Three biological replicates were analyzed for each species.

### 4.3. Exogenous Ethylene Treatment

The triple response assay was carried out to compare variations in ethylene responsiveness among seedlings of *S. lycopersicum* cv. Ailsa Craig, *S. pimpinellifolium*, *S. pennellii*, and *S. peruvianum*. Two day old seedlings were grown on Murashige and Skoog (MS) medium in 50 mL vials, and then treated with 100 ppm of ethylene at 24 °C in the dark for 7 days. Seedlings treated with air were used as the control. Root and shoot length were obtained from at least five measurements for each species. The ethylene responsiveness of fruits of Ailsa Craig at 40 DAP and *S. pimpinellifolium* at 38 DAP, which corresponded to the mature green (MG) stage, was examined. For the green-ripe species, *S. pennellii* at 60 DAP and *S. peruvianum* at 40 DAP were used because these times were equivalent to the MG stage of Ailsa Craig and *S. pimpinellifolium*. Fruits were harvested and then stored in a 270 mL or 50 mL open bottle for 2 h in order to remove wound-induced ethylene effects. Fruits were then incubated with 1000 ppm of ethylene at room temperature for 24 h. Fruits treated with air were used as controls.

### 4.4. DNA Extraction and Sequencing

Tomato fruits at different development stages were homogenized using a Multi-Bead Shocker (YASUI KIKAI). Then, 100 mg of fruit powder was used for genomic DNA extraction. Samples were mixed with 400 mL of DNA extraction buffer that consisted of 250 mM NaCl, 200 mM Tris-HCl, 25 mM EDTA (pH 8.0), and 60 μL of 10% SDS, followed by shaking. Mixed samples were incubated at 65 °C for 30 min. Next, 150 μL of 5 M KAC was put in the mixed samples and stored on ice for 30 min. Phenol/chloroform/isoamyl alcohol (300 μL) was added after incubation in ice and centrifuged at 13,000 rpm for 5 min. The supernatant was collected, before adding an equal amount of 2-propanol. The supernatant/2-propanol mixture was centrifuged for 10 min at 5000 rpm. After centrifugation, the resulting precipitate was washed with 70% ethanol and then resuspended in water. Genomic DNA concentration was measured using a Multiskan spectrophotometer (Thermo Fisher Scientific, Waltham, MA, USA).

The upstream sequence of *S. peruvianum* was amplified with gene-specific primers (Appendix A), and a QIAquick PCR purification kit (QIAGEN) was used for PCR product purification following the manufacturer’s protocol. TA cloning was carried out with the pGEM-T easy Vector System (Promega, Madison, WI, USA). Positive clones were screened through colony PCR with specific primers. QIAprep Spin Miniprep kit (QIAGEN) was used for plasmid DNA extraction following the manufacturer’s protocol. The product was used as the template for sequencing with the BigDyeKit (Applied Biosystems, Inc., Foster City, CA, USA).

### 4.5. Promoter Analysis

Promoter sequences of ethylene biosynthetic genes were downloaded from the National Center for Biotechnology Information (NCBI; https://www.ncbi.nlm.nih.gov/ (accessed on 1 June 2022)) and Sol Genomics Network (https://www.solgenomics.net/ (accessed on 1 June 2022)). Multiple sequences were aligned using the Geneious software. Conserved motifs were determined using MEME 5.3.3 (https://meme-suite.org/meme/tools/meme (accessed on 1 June 2022)) and plotted using TBtools [39]. Regulatory sequence elements were determined using Plant CARE (http://bioinformatics.psb.ugent.be/webtools/plantcare/html/ (accessed on 1 June 2022)). Transposable elements were determined using Repbase (https://www.girinst.org/repbase/update/index.html (accessed on 1 June 2022)).

### 4.6. RNA Extraction and Quantitative RT-PCR Analysis

Firstly, 100 mg of fruit powder was used for total RNA extraction using TRIzol (Invitrogen, Waltham, MA, USA) following the manufacturer’s instructions. At least three biological replications were used. The quality of total RNA was checked through agarose gel electrophoresis using 2× RNA loading buffer (FUJIFILM). Concentrations of total RNA were measured using a Multiskan spectrophotometer (Thermo Fisher Scientific). Complementary DNA (cDNA) was synthesized using ReverTra Ace qPCR RT Master Mix and then used as the template for qRT-PCR. qRT-PCR was performed using a StepOne Plus (Applied Biosystems, Foster City, CA, USA) real-time PCR system. Each reaction contained 5 μL of Fast SYBR Green Master Mix, 10 nM of forward and reverse primers, 2.4 μL of MilliQ water, and 2 μL of template. The qRT-PCR program consisted of the following steps: 95 °C/20 s incubation, 40 cycles of 95 °C/10 s, and 60 °C/30 s. Transcript quantification was dependent on the 2^−ΔΔCt^ analysis method [40]. *Sl-Ubiquitin* was used as the internal reference gene. Student’s *t*-tests were used to detect statistical significances using Microsoft Excel (version 2016).

### 4.7. DNA Methylation Using Real-Time PCR (qAMP)

The DNA methylation level of cultivated tomato was obtained from the tomato epigenome database (http://ted.bti.cornell.edu/epigenome/ (accessed on 5 February 2022)). DNA products were digested using the methylation-sensitive restriction enzyme (MSREs), *Hha*I (TAKARA), for 2 h. Replacement of the enzyme with water was used as the control. Each PCR reaction contained 5 μL of GoTaq Green Master Mix, 10 nM of forward and reverse primers (Appendix A), 0.8 μL of 25 nM MgCl_2_, 1.8 μL of MilliQ water, and 2 μL of digested DNA. For control samples, the PCR program consisted of the following steps: 95 °C/3 min incubation, 35 cycles of 95 °C/30 s, 52 °C/30 s, and 72 °C/30 s. For digested DNA, the PCR program consisted of the following steps: 95 °C/3 min incubation, 40 cycles of 95 °C/30 s, 52 °C/30 s, and 72 °C/30 s. Data were collected from three biological replicates.

## 5. Conclusions

In conclusion, the transcription regulation of ethylene and carotenoid biosynthesis revealed interspecific variations in tomato and related wild species. Activation of *ACS* genes expression in red-ripe tomatoes resulted in increased involvement of a number of *ACS* genes in ripening. The involvement of NOR and RIN in ethylene biosynthesis appears to be an acquired mechanism when the red-ripe tomatoes developed special features to improve its chance of survival. The upregulated gene expression is probably accompanied by the recruitment of activators to several enhancer sites in the upstream regulatory region. Cultivated tomato inherited these features from red-ripe tomato wild relatives in which the enhanced dosage of ACS enzyme caused a burst of ethylene production. In addition to carotenoid biosynthesis, lycopene accumulation is induced by ethylene-mediated regulations, and lycopene breakdown is decreased during ripening in red-ripe tomatoes. By contrast, green-ripe wild relatives lack ethylene production and lycopene accumulation; however, they have a higher lycopene breakdown capacity, which keeps fruits green during maturation.

## Figures and Tables

**Figure 1 ijms-23-10788-f001:**
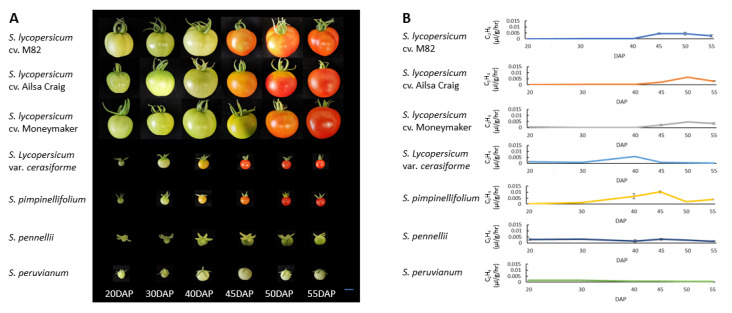
Comparison of fruit ripening and ethylene production among tomato species and cultivars. Changes in skin color (**A**) and quantification of ethylene production during fruit development (**B**). Scale bar = 1 cm. DAP: days after pollination. Fruits were observed from 20 to 55 DAP for each species and cultivar. Error bars in (**B**) represent the standard deviation (SD) of five biological replicates.

**Figure 2 ijms-23-10788-f002:**
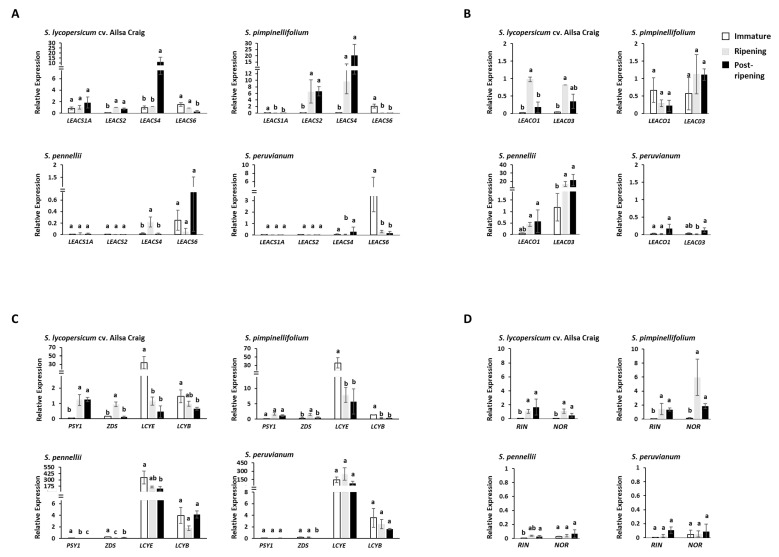
Quantitative RT-PCR analysis of ethylene biosynthetic genes and carotenoid biosynthetic genes. Relative quantification revealing dynamic change in gene expression of *LEACS* genes (**A**), *LEACO* genes (**B**), carotenoid biosynthetic genes (**C**), and the key ripening transcription factors (**D**). The white bars indicate the relative expression levels of genes at the immature stage. Gray bars indicate relative gene expression levels at the start of fruit ripening, and black bars correspond to relative gene expression levels at the over-ripening stage. The gene expression level of *S. lycopersicum* cv. Ailsa Craig at the ripening stage was selected as a reference sample. Significant differences (*p* < 0.05) are indicated by lowercase letters. *ACS*, 1-aminocyclopropane-1-carboxylic acid (ACC) synthase gene; *ACO,* ACC oxidase genes; *PSY1*, *Phytoene synthase 1*; *ZDS*, *z-carotene desaturase**; LCYE*, *lycopene ε-cyclase*; *LCYB**, lycopene β-cyclase*; *RIN*, *Ripening-Inhibitor*; *NOR*, *Non-Ripening*.

**Figure 3 ijms-23-10788-f003:**
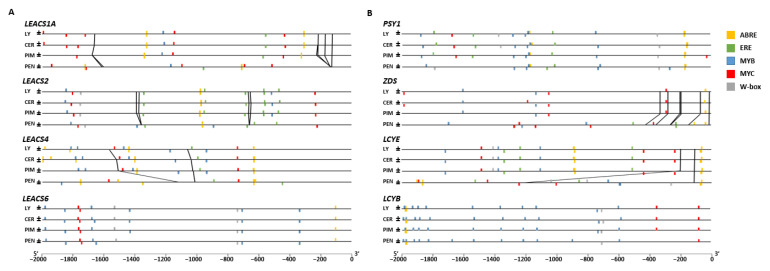
In silico analysis of cis-regulatory elements. Regulatory elements are found at upstream sequences of ethylene biosynthetic genes (**A**) and carotenoid biosynthetic genes (**B**). Insertions and deletions at the regulatory sequences are connected by black lines. Yellow bars, ABA-responsive element (ABRE); green bars, ethylene-responsive element (ERE); blue bars, MYB protein-binding domain (MYB); red bars, MYC family protein-binding domain (MYC); gray bars, WRKY TF-binding domain (W-box). LY, *S. lycopersicum*; CER, *S*. *lycopersicum* var. *cerasiforme*; PIM, *S. pimpinellifolium*; PEN, *S. pennellii*.

**Figure 4 ijms-23-10788-f004:**
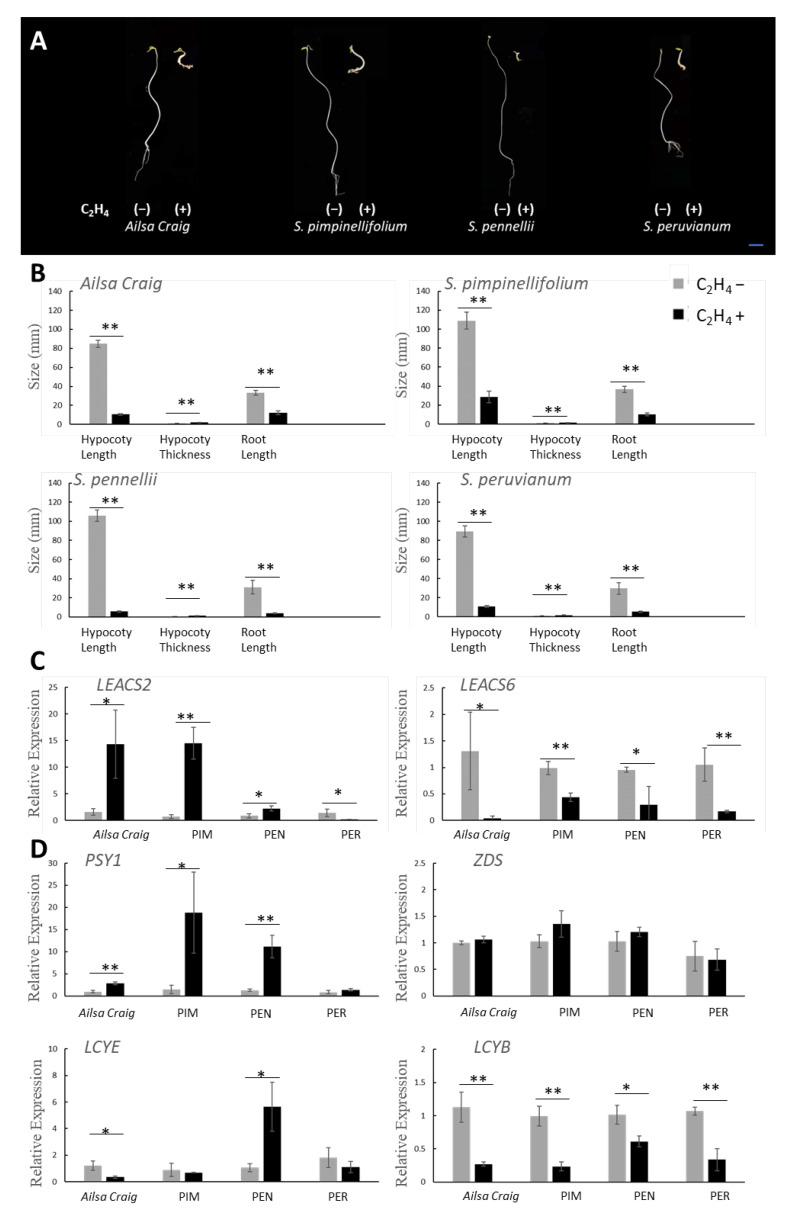
Responsiveness of tomato species to exogenous ethylene. Triple response assay of seedlings (**A**,**B**) and gene expression in fruits exposed to exogenous ethylene (**C**,**D**). Gray bars represent controls, while black bars indicate ethylene-treated samples. *Ailsa Craig*, *S. lycopersicum* cv. Ailsa Craig; PIM, *S. pimpinellifolium*; PEN, *S. pennellii*; PER, *S*. *peruvianum*. Asterisks indicate significant differences as determined by the *t*-test (* *p* < 0.05, ** *p* < 0.01).

**Figure 5 ijms-23-10788-f005:**
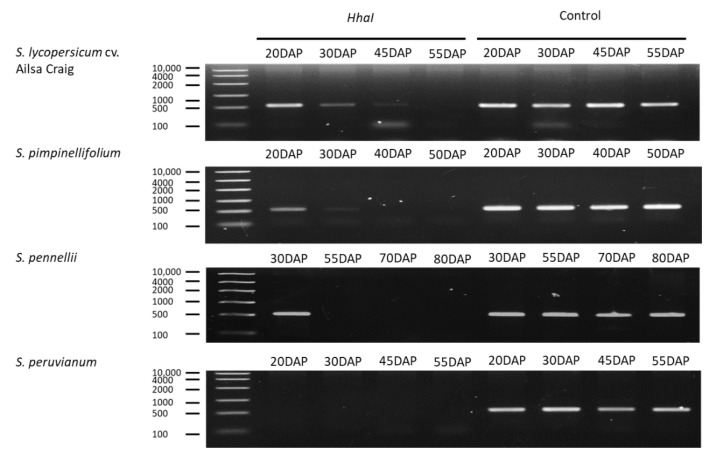
Analysis of DNA methylation by qAMP in upstream regions of *NOR*. Levels of methylation change over time. DAP: days after pollination.

**Figure 6 ijms-23-10788-f006:**
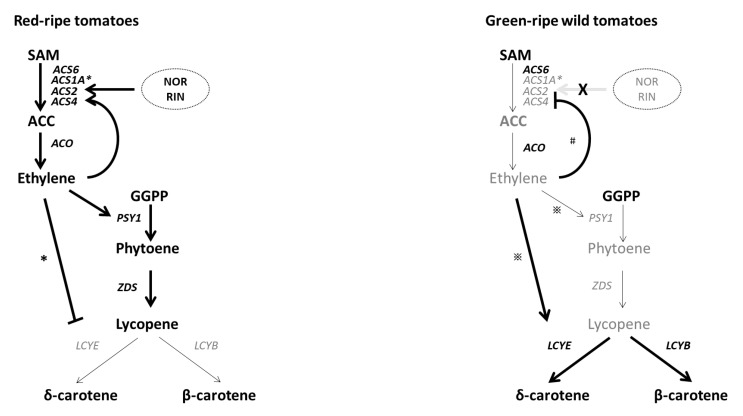
Current understanding of genetic control of ethylene biosynthesis and ethylene-mediated coloration in tomato. Bold lines and thin lines indicate the strength in corresponding activity. Characters in black indicate highly accumulated intermediates or enzymes, and characters in gray indicate relatively lowly accumulated intermediates or enzymes. * Significant response in *S. lycopersicum* cv. Ailsa Craig; ※ significant response in *S. pennellii*; # significant response in *S. peruvianum*.

## Data Availability

Not applicable.

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
