# Peer review of "Increased ACS Enzyme Dosage Causes Initiation of Climacteric Ethylene Production in Tomato"

_ijms, 2022, doi:10.3390/ijms231810788_

Round 1

Reviewer 1 Report

The manuscript by Chen et al. investigate the relation between ACS enzyme production to ethylene production in tomato. The authors shows that increased dosage of ACS enzyme enhances the ethylene production in tomatoes marking climacteric response. 

Major comments: 

  1. The authors should knockdown ACS genes cascade (LEACS2 or LEACS4) to show that ACS enzyme production is directly related to ethylene production in tomatoes to support the major conclusion. 
  2. According to the Fig 2a. the dynamics ACS gene expression is different in Alisa Craig or S. pimpinellifolium, LEASCS1A and LEASCS4 was expressed in Alisa Craig vs. LEACS2 an LEACS4. The authors should provide rational how magnitude of gene expression related to enhanced ethylene production. 
  3. The authors should plot ACS genes, ACO genes, and RIN/NOR in three different plots so that scale is comparable. Also, authors clarify a and b in the figure legend. 
  4. For Fig 2b. the authors should define genes and there function in introduction or result section.
  5. In fig 5, if there is no significant gene expression of NOR in Fig2 of S. pennellii, why it shows hypermethylated NOR promoter.

Author Response

Dear Reviewer:

Thank you for your detailed review and valuable comments. We have carefully revised the manuscript text according to your comments. The changes we have made are highlighted using the “Track Changes” function in the marked revised manuscript. The following part is the point-by-point responses to your comment. The number of line indicates the line in “Track Changes” function.

Reviewer’s comment 1

The authors should knockdown ACS genes cascade (LEACS2 or LEACS4) to show that ACS enzyme production is directly related to ethylene production in tomatoes to support the major conclusion.

<Response>

Oeller et al. (1991) have demonstrated that LEACS2 knockdown fruits resulted inhibition of ethylene production, and Hoogstrate et al. (2014) demonstrated that a truncated protein inhibited LEACS4 mutant reduced ethylene production in cultivated tomato (Solanum lycopersicum var. TPAADASU). We added information in the revised manuscript at line 77 to line 78.

Reviewer’s comment 2

According to the Fig 2a. the dynamics ACS gene expression is different in Alisa Craig or S. pimpinellifolium, LEASCS1A and LEASCS4 was expressed in Alisa Craig vs. LEACS2 and LEACS4. The authors should provide rational how magnitude of gene expression related to enhanced ethylene production.

<Response>

We confirmed that ethylene evolution is higher in S. pimpinellifolium (figure1B) and the maximum value has described in results (line 119 to 126) which is identical to your suggestion. ACS is rate-limiting enzyme that higher gene expression (LEACS2 and LEACS4) correspond to enhanced ethylene production. We have added this information to the revised manuscript, line 358 to 362.

Reviewer’s comment 3

The authors should plot ACS genes, ACO genes, and RIN and NOR in three different plots so that scale is comparable. Also, authors clarify a and b in the figure legend.

<Response>

We have revised Fig 2 based on your kind suggestion. Please find the revised· Fig 2 and legends at line 170 to 181. In addition, we have revised manuscript according to new Fig 2. Please find the text from line 160 to 167.

Reviewer’s comment 4

For Fig 2b. the authors should define genes and their function in introduction or result section.

<Response>

 We have added this information to the revised manuscript, line 155 to 160.

Reviewer’s comment 5

In fig 5, if there is no significant gene expression of NOR in Fig2 of S. pennellii, why it shows hypermethylated NOR promoter.

<Response>

In this study, we opine that the NOR-mediated ethylene production has not been ruled out in green ripe cultivars. Zhong et al. (2013) have demonstrated a dynamic regulation of NOR gene expression by epigenetic demethylation in ripening state. The hypermethylated NOR promoter observed at immature stage indicate that epigenetic regulation of NOR has been established. Although S. pennellii and S. peruvianum are green ripe, the genetic architecture of gene expression regulation is highly diversified among plants.

Reviewer 2 Report

REVIEW REPORT

The manuscript entitled “Increased ACS Enzyme Dosage Causes Initiation of Climacteric Ethylene Production in Tomato” represents a valuable study on the bursts of ethylene production that accompanies fruit color changes in red-ripe tomatoes and is likely to be an evolutionary adaptation for seed dispersal. The aim of this study was to examine the effects of increased dosage of ACS enzyme in different tomato cultivars. The study showed that the expression level of ACS enzymes was higher in red-ripe tomato fruits than in green-ripe tomato fruits cultivars. The increased ACS enzyme dosage stimulated ethylene production, which corresponded to the climacteric response of red-ripe tomato fruits.

The manuscript is written well, but needs some corrections as follows.

Introduction

The introduction section is well written. The concern is to add all the latest and relevant published literature.

Line 13: Oxidase should be written as “oxidase”.

Line 20: First define ACS1A

Line 33: What is the meaning of “section”?

Line 57-60: These lines are repeated. If possible, write in short or one sentence.

Results

Line 98: First define “DAP” in the text.

Add the reason for studying expression profiling only with S. lycopersicum, cv. Alisa Craig and not with the other cv. of S. lycopersicum.

Figure 2 Visualization is very poor. The white bars are not clear. Text written in the graph was unable to be read. Improve Figure 2 and use bold letters. Abbreviations of the graph should be defined in the figure legend.

Line 151: define ZDS

If possible, add a few data on the biochemical assay before doing molecular level.

Discussion

Discussion should be improved. The discussion part is not related to the actual problem studied. It was general. Discussion lacks theme/title flow. Discussion should be linked in a better way to the other parts of the manuscript. With the additional interpretation of the data and a better explanation, the discussion may be more appealing and intriguing.

Additionally, in the discussion section, first describe the problem studied, and the novelty of the work then go through the discussion/reference part in another paragraph. Relate with other studies.

Figure 6  Green-ripe wild tomatoes; the step including SAM, ACC, and Ethylene should be bold as Red–ripe tomatoes.

Material and Methods

What is the meaning of “? I” in the equation in section 4.2?  There is no need to write FW after fresh weight in the equation. It may be written after “where”.

Check format style in section 4.4. There is some error in the journal format style.

Line 471: SlUbiquitin should be Sl-Ubiquitin

The Conclusion section is missing. The important points of the manuscript should be summarized in the conclusion.

References must be formatted in accordance with the journal's reference style. Standardize the references.

Author Response

 Dear reviewer,

Thank you for your detailed review and valuable comments. We have carefully revised the manuscript text according to your comments. The changes we have made are highlighted using the “Track Changes” function in the marked revised manuscript. The following part is the point-by-point responses to your comment. The number of line indicates the line in “Track Changes” function.

Reviewer’s comment

Introduction

Line 13: Oxidase should be written as “oxidase”.

<Response>

We have revised manuscript according to your comment. Please find the text in line 13.

Line 20: First define ACS1A

<Response>

We have revised manuscript according to your comment. Please find the text in line 20.

Line 33: What is the meaning of “section”?

<Response>

Section indicates taxonomic rank below the genus but above the species. Solanum section Lycopersicom is used to define the plant group including the cultivated tomato (S. lycopersicum) and 12 additional wild relatives.

Line 57-60: These lines are repeated. If possible, write in short or one sentence.

<Response>

We have revised manuscript according to your comment. Please find the text from line 59 to 61.

Results

Line 98: First define “DAP” in the text.

<Response>

We have revised manuscript according to your comment. Please find the text in line 102.

Add the reason for studying expression profiling only with S. lycopersicum, cv. Alisa Craig and not with the other cv. of S. lycopersicum.

<Response>

We have revised manuscript according to your comment. Please find the text from line 137 to 139.

Figure 2 Visualization is very poor. The white bars are not clear. Text written in the graph was unable to be read. Improve Figure 2 and use bold letters. Abbreviations of the graph should be defined in the figure legend.

<Response>

The Figure 2A is divided into three according to your and the other reviewer’s comment. The expression level of ACS genes, ACO genes, and TFs are showed in different plot. Abbreviations have been defined in the new figure legend.

Line 151: define ZDS

<Response>

We have revised manuscript according to your comment. Please find the text from line 157 to 158.

If possible, add a few data on the biochemical assay before doing molecular level.

<Response>

We are currently working towards setting up a biochemical assay, especially in wild tomatoes. We are regret that we cannot provide data immediately, and hope to update the data in subsequent research papers.

Discussion

Discussion should be improved. The discussion part is not related to the actual problem studied. It was general. Discussion lacks theme/title flow. Discussion should be linked in a better way to the other parts of the manuscript. With the additional interpretation of the data and a better explanation, the discussion may be more appealing and intriguing.

Additionally, in the discussion section, first describe the problem studied, and the novelty of the work then go through the discussion/reference part in another paragraph. Relate with other studies.

<Response>

Thanks for your suggestion. We have revised based on your comment.

Figure 6  Green-ripe wild tomatoes; the step including SAM, ACC, and Ethylene should be bold as Red–ripe tomatoes.

<Response>

Thanks for your suggestion. Just to confirm, we attempted to reveal quantitative differences using gray characters in Figure 6. Our survey found lower level ethylene production in green-ripe species (Figure 1B). Although we could not measure the concentration of ACC, relative low level of ACS genes expression (Figure 2A) may reveal a lower concentration.

Material and Methods

What is the meaning of “? I” in the equation in section 4.2?  There is no need to write FW after fresh weight in the equation. It may be written after “where”.

<Response>

Thanks for your notification, we are very sorry for our incorrect writing. The “I” was a mistake, it should be μl. We have revised manuscript according to your comment. Please find the text from line 466.

Check format style in section 4.4. There is some error in the journal format style.

<Response>

Thanks for your suggestion. We have revised based on your comment. Please find the text in line 454 and line 487.

Line 471: SlUbiquitin should be Sl-Ubiquitin

<Response>

We have revised manuscript according to your comment. Please find the text in line 524.

The Conclusion section is missing. The important points of the manuscript should be summarized in the conclusion.

<Response>

 We have added conclusions following Materials and Methods section.

References must be formatted in accordance with the journal's reference style. Standardize the references.

<Response>

We have revised manuscript according to your comment.

Round 2

Reviewer 1 Report

All queries were answered.

Reviewer 2 Report

The manuscript has been revised according to my suggestions. It is now acceptable.